# Position: Adversarial ML for LLMs Is Not Making Any Progress

## Abstract

In the past decade, considerable research effort has been devoted to securing machine learning (ML) models that operate in adversarial settings. Yet, progress has been slow even for simple "toy" problems (e.g., robustness to small adversarial perturbations) and is often hindered by non-rigorous evaluations. Today, adversarial ML research has shifted towards studying larger, general-purpose language models. In this position paper, we argue that the situation is now even worse: **in the era of LLMs, the field of adversarial ML studies problems that are (1) less clearly defined, (2) harder to solve, and (3) even more challenging to evaluate.** As a result, we caution that yet another decade of work on adversarial ML may be failing to produce meaningful progress.

## 1 Introduction

When adversarial machine learning emerged as a field, it focused on attacking and defending simple models with well-defined objectives. For example, misclassifying a spam message as safe (Graham-Cumming, 2004) or images in deep learning models (Biggio et al., 2013; Szegedy, 2013; Goodfellow et al., 2014). These early problems were well-defined: the attack goals were clear (e.g., cause a misclassification), the target models were relatively simple (e.g., linear classifiers, small neural networks), the threat models were simple (e.g., perturb pixels by at most 8/255), and the evaluation metrics were straightforward (e.g., accuracy on a test set). Yet the field has struggled to develop robust solutions or even to fully understand why these vulnerabilities exist (Barreno et al., 2006; Shafahi et al., 2019). Even fundamental "toy" problems like robustness to $\ell_p$-bounded perturbations, remain largely unsolved to this day, and many defense evaluations still lack rigor (Carlini & Wagner, 2017; Carlini et al., 2019; Tramer et al., 2020).

Recently, the focus of the field has since shifted towards studying adversarial problems with large language models (LLMs) and other generative models. **In this position paper, we argue that these new problems are significantly harder to define, solve and evaluate; making progress increasingly difficult to track**.

Due to their general-purpose nature, LLMs are not designed to solve any single well-defined "task" to be secured. Instead, the field now considers a more holistic notion of "safety", with adversarial objectives that are hard to define formally (e.g., making an LLM produce "harmful" responses) (Christiano et al., 2017; Ouyang et al., 2022; Bai et al., 2022; Casper et al., 2023). These safety properties are also often considered for unbounded threat models, thereby leading to much stronger adversaries (e.g., with the ability to adversarially fine-tune a model or to prompt it in arbitrary ways). Due to this large attack space—and the difficulty of directly optimizing over it (Carlini et al., 2024)—attacks are increasingly ad-hoc and human driven (Li et al., 2024a). This further complicates the task for defenders, who cannot automatically search over strong, adaptive attacks.

Submitted to 39th Conference on Neural Information Processing Systems (NeurIPS 2025). Do not distribute.

Table 1: Challenges in different research areas when defining and solving adversarial ML problems.

| | Challenges | | | | | | |
| | Defining | | | Solving | | Evaluating | |
| Research Area | (§2.1.1) Defining Success | (§2.1.2) Bounding Attacks | (§2.1.3) Delimiting Data | (§2.2.1) Attack Search | (§2.2.2) Principled Defenses | (§2.3.1) Measuring Harm & Utility | (§2.3.2) Ensuring Reproduc. |
|---|---|---|---|---|---|---|---|
| (§3.1) Jailbreaks | ✓ | ✓ | | ✓ | ✓ | ✓ | ✓ |
| (§3.2) Un-finetunable Models | ✓ | ✓ | | ✓ | ✓ | ✓ | |
| (§3.3) Poisoning + Backdoors | ✓ | ✓ | ✓ | ✓ | ✓ | ✓ | ✓ |
| (§3.4) Prompt Injections | ✓ | ✓ | | ✓ | ✓ | ✓ | ✓ |
| (§3.5) Membership Inference | ✓ | | ✓ | | | | ✓ |
| (§3.6) Unlearning | ✓ | ✓ | ✓ | ✓ | ✓ | ✓ | |

Beyond making the technical problems harder, we argue that generative models have also made evaluation and benchmarking of attacks and defenses more challenging. Measuring attack success is no longer as straightforward as measuring misclassification rates; it instead requires careful (human) evaluation of possible harms present in natural language outputs (Mazeika et al., 2024; Chao et al., 2024). In a similar vein, evaluating whether defenses preserve the utility of the original model has become more nuanced: instead of measuring test accuracy on a single task, we now have to determine whether a model maintains its general-purpose capabilities (Cui et al., 2024; Mai et al., 2025).

Finally, reproducible benchmarking became harder as many state-of-the-art models are deployed via black-box APIs that may receive constant updates and patches as newer attacks are released. As these changes are often not reported, reproducing results or making meaningful comparisons between different approaches becomes nearly impossible.

In this position paper, we use several case studies of research areas in adversarial ML to illustrate the increasing complexity in both attacks and defenses. We first analyze how traditional research problems have evolved to become fundamentally harder to formally define and solve (Section 2). We then present case studies that illustrate these new challenges (Section 3). Finally, we discuss our perspective on why these changes represent a fundamental challenge to progress in the field and alternative views on the evolution of adversarial ML (Section 4).

## 2 New Challenges in Defining, Solving, and Evaluating Adversarial ML Problems

Traditional ML models were designed and trained for specific and narrow tasks—often classification. For example, computer vision models used to classify images into a fixed set of classes (Krizhevsky et al., 2012), and natural language processing models used to perform textual analysis on individual sentences (Richardson et al., 2013; Rajpurkar, 2016). Additionally, the training and test data were clearly delineated as inputs were discrete and bounded units (individual images or sentences). In these settings, adversarial objectives could be clearly specified. For example, misclassifying as many inputs as possible (i.e., adversarial examples (Szegedy, 2013; Goodfellow et al., 2014)) or inferring if a given data point was used for training (i.e., membership inference (Shokri et al., 2017)).

However, LLMs have fundamentally changed this landscape. Models no longer perform narrow tasks but serve as general-purpose systems that produce free-form and unbounded outputs. As a result, defining "security" or "safety" properties of the AI system has become more challenging, with the field focusing on general definitions (e.g., a model should not produce outputs that can "harm others"[1]). Adversarial objectives related to training data (e.g., membership inference or unlearning) have also become more ill-defined, as the training set(s) of LLMs span virtually the entire Internet (Gao et al., 2020), with no clear boundaries between data points or between train and test sets.

In this section, we identify three core challenges, each split into several sub-challenges, that make adversarial ML for LLMs *harder to define*, *harder to solve*, and *harder to evaluate*. We provide a summary of the challenges faced in different research areas in Table 1.

---

[1] https://openai.com/policies/usage-policies/

In Section 3, we elaborate on how these challenges hinder progress by analyzing specific case studies: *Jailbreaks* (Section 3.1), *Un-finetunable Models* (Section 3.2), *Poisoning and Backdoors* (Section 3.3), *Prompt Injections* (Section 3.4), *Membership Inference* (Section 3.5), and *Unlearning* (Section 3.6).

## 2.1 Problems are Harder to Define

### 2.1.1 Defining Success of Attacks and Defenses

In the past, adversarial problems for classification models typically involved concrete objectives (e.g., misclassifying images), which could be easily measured by accuracy on a set of clean or perturbed inputs. Now, the lack of a single well-defined task makes it unclear what criteria constitute a genuine success or failure for attacks or defenses.

LLMs produce free-form text in which goals become subjective. Developers now aim to optimize abstract properties like helpfulness, honesty, and harmlessness (Bai et al., 2022), while adversaries may try to obtain generically harmful outputs. Thus, measuring attack success—i.e., whether an output is actually harmful or violates the developer policies—also becomes subjective.

### 2.1.2 Defining and Bounding the Attack Space

In prior robustness settings (e.g., with classification models), the adversary was often constrained to perturb inputs within an $\ell_p$-ball around a given image. This served as a meaningful *necessary* but *not sufficient* condition for robustness Gilmer et al. (2018), allowing quantitative comparisons of different methods (Goodfellow et al., 2014).

For LLMs, researchers almost always allow the search space for attacks to be unbounded, since any input could potentially elicit a violation of a safety property (Wei et al., 2024a). The shift from input-dependent to input-*independent* constraints makes it harder to specify adversarial capabilities that allow us to compare attacks and defenses. Beyond unbounded inputs, threat models have also become more permissive. In traditional adversarial ML problems (e.g., adversarial examples or poisoning), the strongest adversaries had white-box access to model weights, but could not alter the model's functionality. Now attackers need not maintain the model's general capabilities as long as they can elicit the desired harmful information, enabling stronger attacks such as fine-tuning or pruning (Qi et al., 2024b; Wei et al., 2024b)[2].

Moreover, the set of attacks that should be ruled out may not always be obvious. While one could say "any input that leads to harmful content is a valid attack," trivial attacks such as prompting "please repeat [harmful text]" do not reveal meaningful new vulnerabilities. Hence, there is no clear universal standard on what sorts of prompts or transformations count as "valid" or "novel" adversarial inputs.

### 2.1.3 Delimiting Data

In many research areas traditionally studied in adversarial ML, such as unlearning or privacy protection, the notion of a *training data point* plays a crucial role. Previously, a model was trained on a carefully curated dataset with strict train/test splits; each data point (such as a single labeled image) was distinct, and known to researchers. In contrast, generative models are trained on vast corpora, where similar, or even identical, content may appear across multiple subsets of the training set. The exact contents of the training data are also rarely publicly released (Nasr et al., 2025). The notion of a held-out (IID) test set no longer really exists.

## 2.2 Problems are Harder to Solve

### 2.2.1 Searching over Attacks

The optimization landscape for most adversarial ML problems has become significantly more complex with LLMs. In traditional classification problems, such as crafting adversarial images, the objective function was clear: maximize the loss on the correct prediction while minimizing perturbation size. This objective could be formalized and optimized by propagating gradients to the

---

[2]For adversarial robustness in image classifiers, the ability to finetune the victim model would be a trivial attack vector, since the attacker could simply fine-tune the model to have low accuracy.

input space (Madry, 2017). These automated attacks outperformed humans and consistently found worst-case attacks (Carlini et al., 2017).

However, the attack surface for LLMs is much larger and harder to define (see Section 2.1.2). There is no longer a single well-defined "task", and safety properties cannot be expressed with formal loss functions—they are qualitative, context-dependent, and often subjective (Bai et al., 2022).

Even if we define a "toy" attack objective (e.g., making the model output an affirmative response such as "Sure, I can help you with that" (Zou et al., 2023)), finding good attacks remains hard (Carlini et al., 2024). Discrete text inputs makes gradient-based methods less effective (Carlini et al., 2024; Rando et al., 2024), and the vast search space makes exploration impractical. Perhaps most telling, manual attacks still outperform automated methods at finding worst-case inputs (Li et al., 2024a). Many successful attacks on LLMs exploit qualitative properties that are hard to optimize automatically, such as persona modulation (Shah et al., 2023), multi-turn conversations (Anil et al., 2024), and social engineering techniques (Zeng et al., 2024). In contrast, current optimization methods typically generate gibberish inputs (Zou et al., 2023; Thompson & Sklar, 2024).

### 2.2.2 Building Principled Defenses

In traditional adversarial tasks, researchers could devise *certified* defenses (Cohen et al., 2019) or well-motivated empirical defenses such as adversarial training (Madry, 2017), where key properties of the problem (like bounded input perturbations) were explicitly understood. Moreover, the performance of these defenses could be evaluated with strong, adaptive white-box attacks (Tramer et al., 2020).

In contrast, for LLMs the adversarial objectives are typically not formally defined (see Section 2.1.1) and the attack space is challenging to bound (see Section 2.1.2). As a result, there is little hope to build defenses upon principled foundations. Existing defenses rely on ad-hoc approaches, through either: (1) adversarial training against *known* successful attacks Bai et al. (2022); Wallace et al. (2024); (2) "virtual" adversarial training in the model's latent space Miyato et al. (2018); Casper et al. (2024b); Sheshadri et al. (2024); (3) building external classifiers or detectors (Inan et al., 2023); (4) or random preprocessing (Robey et al., 2023). Crucially, none of these approaches produce systems whose security can be analyzed or quantified in a well-defined formal. It is thus not too surprising that the original evaluations of some of these defenses overestimate their robustness (Chi et al., 2024; Qi et al., 2024a; Łucki et al., 2024).

## 2.3 Problems are Harder to Evaluate

### 2.3.1 Measuring Attack Harm and Defense Utility

Since safety properties for LLMs are hard to formally define, it has become customary to use LLMs themselves as a fuzzy "judge" to determine harmfulness (e.g., when evaluating jailbreaks or prompt injections (Mazeika et al., 2024)). But this approach suffers from a number of issues. First, judges fall short of human judgment.[3] For instance, many implementations often default to considering any non-refusal response as a successful attack even if the content is harmless (Souly et al., 2024). Second, judges themselves may be vulnerable to attacks (Mangaokar et al., 2024; Raina et al., 2024). Third, using LLMs-as-judges to evaluate defenses can create artificial correlations that bias evaluations. For example, a defense that implements an output filter similar to the judge may achieve near-perfect scores without necessarily being effective against prompts where the judge fails (Liu et al., 2024).

Measuring benign utility of defenses—whether they preserve other capabilities—is also non-trivial. Unlike classification tasks where accuracy on a fixed test set is standard, LLMs can be used for an open-ended array of tasks. A defense can trivially produce a safe-but-useless model by refusing all requests. Thus, any evaluation framework must somehow account for the model's usefulness to the end-user, which is subjective and context-dependent (Cui et al., 2024).

---

[3]Even (non-expert) humans have a hard time judging harmfulness of model responses, e.g., when judging whether "instructions for building a bomb" truly yield a useful design.

### 2.3.2 Reproducing and Comparing Results

In earlier, more controlled research environments, practitioners had detailed information about a model's architecture, training data, and training pipeline, enabling precise definitions of threats, defenses, and success criteria. This transparency made it straightforward to track progress.

Many influential LLMs are now closed-source and updated silently over time (Chao et al., 2024), making it unclear which version of a system is being tested. Moreover, instead of investigating a single, well-defined model, one must analyze an entire system that may incorporate multiple pre-processing, post-processing, or other defense mechanisms.

This lack of transparency severely undermines reproducibility. Researchers cannot confirm whether observed behaviors persist across different snapshots of the system, nor can they reliably benchmark potential solutions. Consequently, adversarial ML problems become harder to define—let alone solve and evaluate. While black-box or discrete optimization approaches can help reveal some vulnerabilities, they provide only limited insight into the model's internals, leaving many critical security and privacy questions unanswered (Casper et al., 2024a; Carlini et al., 2024).

## 3 Case Studies

### 3.1 Jailbreaks

Jailbreaks illustrate many of the new challenges in adversarial research. Jailbreaks are adversarial text inputs for language models that bypass safeguards to generate "harmful" content (Wei et al., 2024a).

**"Harmful" content has no formal definition.** Defining success for an adversarial image is relatively easy: the perturbation is "small" under some given measure, and leads to a misclassification. With jailbreaks, however, success requires defining what it means for a model to output "harmful" or otherwise "undesirable" content. Early attempts used crude proxies based on simple substring matching (Zou et al., 2023). This approach has largely been replaced by a more general use of an "LLM-as-a-judge", where the fuzzy task of defining harmfulness is given to another LLM (Zheng et al., 2023; Chao et al., 2023; Shah et al., 2023; Mazeika et al., 2024). The circularity of this definition leads to a number of issues, as illustrated in Section 2.

**There are no meaningful bounds on adversaries.** Although adversaries for image classification could also be unbounded, the fact that the safety property is dependent on the input (replacing a cat by a dog is not an interesting attack) made the community define an $l_p$ norm around the inputs as a proxy for preserving visual similarity. However, for jailbreaks, there is not such a meaningful bound as the safety property is *independent* of the input (harmful generations should never occur). Researchers have come up with attacks that use semantic augmentations (e.g., role-playing or social engineering) (Shah et al., 2023; Zeng et al., 2024), append high-perplexity suffixes (Zou et al., 2023; Thompson & Sklar, 2024) or even found that long inputs and random augmentations dilute safeguards (Anil et al., 2024; Andriushchenko et al., 2024; Hughes et al., 2024). Not only adversaries are now unbounded in the input space, but they can use additional methods such as fine-tuning (Qi et al., 2024b) or pruning (Wei et al., 2024b). This diversity of attacks illustrates the difficulty to define a narrow task, analogous to $\ell_p$ bounded robustness, that can be used to compare and benchmark attacks and defenses.

**Optimizing for worst-case attacks is hard.** Optimizing attacks against classifiers is straightforward. You can set as objective the maximization of the model loss (Szegedy, 2013). The loss gradient can be propagated all the way to the input to guide updates. However, LLMs do not provide any of the above: the optimization goal is unclear and optimization is not continuous nor over a finite input space. As a workaround, previous work has tried to optimize proxy objectives such as maximizing the probability of a compliance prefix (e.g. "Sure, I can help you with that") (Zou et al., 2023; Carlini et al., 2024). However, the input space is still discrete and virtually infinite. These challenges make discrete optimization extremely inefficient and close to random search (Zou et al., 2023; Andriushchenko et al., 2024). Optimization challenges have made us shift from a field where the strongest attacks were found via white-box optimization, to one where the best attacks often come from human experts and cannot be found via optimization (Li et al., 2024a). This challenges our ability to make progress in measuring worst-case performance of systems (Carlini et al., 2024).

## 3.2 Unfinetunable Models

A recent research direction aims to design models that are not only robust to jailbreaks, but *also are robust to fine-tuning* Tamirisa et al. (2024); Rosati et al. (2024). This threat model is motivated by the general observation that if a model does *not* have the knowledge to perform some dangerous capability (such as giving instructions for how to perform a cyberattack or design a bioweapon), attacks will never be successful (Li et al., 2024b).

**The attacker is strictly more powerful than for adversarial examples.** An adversarial example attacker has exactly one ability: to modify the input so the model produces an incorrect output. When designing an un-finetunable model, we assume an attacker with *strictly* more power: not only can they change the input arbitrarily, but they can also modify the model itself. Indeed, recent work has already shown how the interplay between modifying the input and modifying the parameters can allow attackers to break many recently proposed defenses Qi et al. (2024a).

**The increased attack space makes it more difficult to evaluate.** In the classical adversarial example literature, the evaluator must ensure exactly one thing is true: the input-space gradient is smooth and following it leads to adversarial examples. In contrast, evaluating an unfinetunable model requires that the much higher *parameter-space* gradients are smooth, something often $1000\times$ higher dimensional. Moreover, the number of hyperparmeters in the evaluation increases significantly, introducing even more room for error (Hönig et al., 2024; Qi et al., 2024a).

## 3.3 Poisoning and Backdoors

In poisoning attacks, adversaries modify a model's training data to affect its behavior on specific examples (Huang et al., 2011) or inject backdoors (Gu et al., 2019). The messy datasets and costly training runs for LLMs make the definition, optimization and evaluation of attacks more challenging.

**Attack goals are hard to enumerate and conflict with intended functionality.** In classification models, adversaries injected training examples with specific triggers that correlated with an output label (Gu et al., 2019). However, in generative models, adversaries trigger fuzzy and complex behaviors like producing harmful content or spreading misinformation (Wan et al., 2023; Rando & Tramèr, 2024a; Zhang et al., 2024b). Not only are these behaviors harder to predict and specify formally, but they also fundamentally conflict with the model's intended functionality since the triggered behavior is often universally undesirable and explicitly trained against (Zhang et al., 2024b).

**Attacks can come from multiple training stages and are hard to optimize over.** Traditional machine learning models had a single training stage on the entire dataset. However, LLMs are first pre-trained and then fine-tuned on (curated) data to turn them into helpful and harmless chatbots (Bai et al., 2022). These different training stages have different properties, may enable different attacks, and can overwrite poisoning in previous stages (Anwar et al., 2024; Zhang et al., 2024b). Also, in LLMs there is no longer a good notion of what constitutes an effective poison nor we can optimize over them (Goldblum et al., 2022).

**Experiments with leading models are computationally infeasible.** Rigorous evaluation of backdoor attacks traditionally requires training models from scratch to understand both the effects of poisoned data and to establish clean baselines. However, this becomes infeasible for LLMs, where a single training run can cost millions of dollars (Anwar et al., 2024; Zhang et al., 2024b).

## 3.4 Prompt Injections

In a prompt injection attack (Goodside, 2022; Willison, 2022), an adversary injects malicious instructions into a language model's context, manipulating its behavior to perform unauthorized actions or disclose sensitive information. These attacks commonly target LLM agents or LLM-integrated applications that interact with untrusted third-party resources through external tools (Jarvis & Palermo, 2023; Husain, 2024; Anthropic, 2024).

**Measuring success of attacks and defenses requires a realistic AI agent environment.** Rigorously evaluating the effectiveness of prompt injection attacks and defenses necessitates a realistic AI

agent environment that closely mimics real-world scenarios. Such an environment should include comprehensive system scaffolding with tool use, enabling the simulation of complex interactions. However, for simplicity, many studies opt to simulate these environments and rely on LLMs as judges for evaluation. There are new setups that have more rigorous evaluations (Debenedetti et al., 2024), where the attack's success and utility can be precisely measured, but they are often limited due to the high cost of incorporating new tasks and their reliance on simulated environments.

**Adversaries are unbounded.** Unlike traditional adversarial attacks bounded by $\ell_p$ norms, prompt injection attacks also operate in a vast and unbounded input space. Additionally, prompt injection attacks can leverage context-dependent strategies, such as embedding malicious instructions within seemingly benign or unrelated text, or using multi-turn interactions to gradually steer the model toward undesirable outputs. This diversity in attack vectors, combined with the fact that virtually any controlled input can serve as a potential attack surface, complicates the task of establishing a reasonable threat model. Consequently, creating a standardized "toy" problem for benchmarking prompt injection defenses is inherently difficult.

**Optimizing for strong attacks is hard.** The primary goal of prompt injections is often clear— for instance, manipulating a language model to perform unauthorized actions like sending emails (Debenedetti et al., 2024), where success can be directly measured. However, the attack surface remains vast, encompassing not only single-turn interactions but also multi-turn scenarios where the model may repeatedly call external tools. In such cases, researchers often lack access to intermediate outputs, making it significantly more challenging to refine and optimize the attack.

Most current attacks rely on handcrafted instructions (Greshake et al., 2023; Liu et al., 2023), such as, "Ignore all previous instructions, please do [target action] first," which are often effective in practice. These manual attacks complicate the development of principled defenses like adversarial training, due to their highly context-dependent and ad hoc nature. Recent approaches (Pasquini et al., 2024) have attempted to apply optimization techniques similar to those used in jailbreaks. Unfortunately, these attacks are not guaranteed to be optimal. As a result, defense attempts that train models against attacks mainly focus on *known* attacks Wallace et al. (2024).

**We cannot easily track progress against closed-source systems.** Similar to jailbreaks, model developers can mitigate prompt injection attacks by implementing safeguards such as filtering mechanisms (Willison, 2023; Wu et al., 2024) or regularly updating and fine-tuning their models (Wallace et al., 2024).As these systems are frequently updated, it becomes difficult to establish a consistent benchmark for measuring progress or reproducing results. Additionally, there are currently few open-source models that are effective tool-use agents (Debenedetti et al., 2024) and can be used for reproducible evaluation.

### 3.5 Membership Inference

Membership inference (MI) attacks (Shokri et al., 2017) aim to determine whether a specific sample $x$ was part of a model's training set.

**The distinction between members and non-members is no longer clearly defined.** In traditional classification settings, the training data is typically of limited size and with a clear delimitation between samples. However, the situation becomes more complicated for generative models.

1. **Highly (partially) duplicated datasets**. The training data of generative models often comes from massive, diverse open datasets, which could include numerous duplicate and near-duplicate samples (Lee et al., 2022; Tirumala et al., 2023). Even if a model appears to memorize a particular sample (e.g., a piece of text or image), this does not necessarily prove that this sample itself was used during training. For example, a model might know much of the plot of Harry Potter without having been explicitly trained on the original book; it could have learned about the story indirectly through Wikipedia pages, reviews, etc. Thus, the boundaries between members and non-members are blurred by the sheer scale and overlap of these datasets.

2. **No IID train and test splits available**. Methods for evaluating MI designate the training data as members and separate IID held-out data as non-members. However, for most generative models, the training datasets are typically not disclosed. Some recent studies attempt to collect non-members

post hoc for evaluation purposes (Shi et al., 2023; Meeus et al., 2023), but these efforts often violate the IID assumption and lead to misleading conclusions (Duan et al., 2024; Das et al., 2024).

**We cannot build counterfactual scenarios for evaluation.**  In traditional classification tasks (e.g., CIFAR-10), where the data generation process is known and models are relatively small, counterfactual scenarios can be built by retraining the same model while excluding a sample $x$, and then comparing statistical behaviors on $x$ (Carlini et al., 2022). In the context of generative models, this approach is ill-defined and computationally impractical, thus it's infeasible to properly evaluate the success of a MI attack (Zhang et al., 2024a).

## 3.6   Machine Unlearning

Machine unlearning was originally formulated as a well-defined task: completely removing the influence of a specific datapoint $x$ from a model (Bourtoule et al., 2021). The goal was to produce a model that, after unlearning $x$, would be indistinguishable from one that was never trained on that point. In traditional classification settings with bounded inputs and outputs, and (often) deduplicated datasets with clear train-test splits, this objective could be precisely defined and evaluated. In fact, there exist exact solutions to unlearning (Bourtoule et al., 2021).

**Unlearning of "concepts" rather than individual data points is hard to define.**  However, generative models have fundamentally changed the nature of unlearning (Cooper et al., 2024). Instead of removing the influence of specific data points, the goal is to remove knowledge about entire concepts or topics that may be contained in one *or more* data points (e.g., all dangerous knowledge about bioweapons (Li et al., 2024b) or copyrighted content from Harry Potter books (Eldan & Russinovich, 2023)). This has made it impossible to define unlearning in terms of a specific data point's influence, making both solutions and evaluations much more challenging.

**Unlearning goals conflict with other knowledge.**  Developers may need to remove very specific knowledge (e.g., bioweapons) while maintaining the model's expertise in related fields (e.g., biology and virology) (Li et al., 2024b). This tension between harmful and benign knowledge makes it inherently hard to define the goal of unlearning and to robustly evaluate safety and utility.

**Threat models are overly strong.**  Unlearning emerged as a white-box protection that would prevent *any* adversary from accessing undesired capabilities (Li et al., 2024b). This ambitious goal also enables stronger threat models where adversaries cannot only query the model, but also finetune it (Hu et al., 2024) and perform any white-box interventions (Łucki et al., 2024). Protecting against such a large attack surface is much harder (Qi et al., 2024a) as discussed in Section 3.2.

**Measuring unlearning success is hard.**  Measuring unlearning success has become significantly more challenging: training baseline models without specific datapoints is costly (Eldan & Russinovich, 2023) and membership inference has important limitations (see Section 3.5). Recent studies have also demonstrated that even when a model cannot generate specific information, this does not reliably prove the underlying knowledge has been erased from its weights (Patil et al., 2023; Lynch et al., 2024; Łucki et al., 2024; Shumailov et al., 2024). In practice, the search for adaptive evaluations is impractical and requires very careful tuning of the methodology for each scenario (Łucki et al., 2024; Qi et al., 2024a). Finally, Shi et al. (2024) showed that measuring unintended effects of unlearning is challenging, as it can significantly affect other capabilities or even amplify privacy leakage.

# 4   Discussion

## 4.1   Alternative Views

**We are solving the right problem in the first place.**  We see increased complexity in adversarial ML because we are finally attempting to solve *real* security challenges rather than toy academic problems. We knew that $\ell_p$-bounded perturbations were a simplified proxy (Gilmer et al., 2018), but they were studied because they were challenging enough to drive progress and served as a *necessary* condition for real-world robustness. We could similarly define toy problems for LLMs (e.g., jailbreaks limited to fixed-length prefixes or bounded sentence modifications), but the field has largely avoided

such artificial constraints in favor of studying real-world unbounded adversaries. This shift might not indicate that problems have become fundamentally harder, but rather that the research community has decided to directly tackle the full complexity of real-world security.

**Solving jailbreaks might be easier because we only need to prevent a behavior regardless of context.** Some researchers argue that certain problems have become simpler with LLMs. For instance, unlike adversarial examples where a model should maintain correct predictions in appropriate contexts (e.g., classify guacamole images as guacamole, but never cats as guacamole), jailbreak prevention has a simpler goal: the model should *never* produce certain harmful outputs (e.g., instructions for building explosives) regardless of context. However, since there are many ways to express this knowledge (e.g., harmful requests can be decomposed into benign subquestions (Glukhov et al., 2024)), defining and evaluating whether a model will *never* produce harmful outputs remains a challenging problem.

Recent work, on representation engineering (Arditi et al., 2024; Zou et al., 2024; Tamirisa et al., 2024) has aimed to identify specific directions in the model's representation space that can anticipate undesired behavior and prevent it universally. Yet, we know that adversarial images could also be detected by similar methods (Carlini & Wagner, 2017), but these defenses ultimately proved vulnerable to newer attacks. Similarly, there are already works that show that representation engineering methods cannot robustly void undesired behaviors (Li et al., 2024a; Qi et al., 2024a).

**Scaffolding to reduce the probability of failure might be sufficient.** Given the difficulty of achieving robust safety guarantees, researchers and companies increasingly rely on complex defense systems (Sharma et al., 2025) and security through obscurity (Rando & Tramèr, 2024b) to minimize risks. While this approach has demonstrated clear benefits in protecting users from harmful content, it prevents rigorous, reproducible and adaptive evaluations as systems become more complex and opaque (Casper et al., 2024a). This trend is particularly concerning given historical lessons: preventing researchers from thoroughly analyzing systems can lead to severe real-world security breaches (Swire, 2004; Mulligan & Perzanowski, 2007; Payne & Parks, 2020). The apparent safety gains from obscurity and complexity may come at the cost of genuine security understanding.

**We are already making progress on these problems.** A prevalent view in the field suggests that we are advancing security capabilities, pointing to newer models being demonstrably harder to attack than their predecessors (Achiam et al., 2023; Zaremba et al., 2025). While this observation might hold generally true, we caution that our inability to robustly evaluate defenses may be hindering our ability to track progress (see Section 2.3). Moreover, we must distinguish between progress in preventing average-case vulnerabilities and achieving **worst-case** security robustness. Although we might be making progress in the former, we have barely improved the latter and most models can still produce harmful generations under attacks. As the stakes increase with more capable models, the risks of rare yet successful attacks become significant (Anthropic, 2023).

## 4.2 Suggestions for improvement

We propose that there are (at least) two valid reasons for performing research on adversarial machine learning: (a) studying real-world security vulnerabilities and (b) advancing scientific understanding of adversarial ML. Papers should be explicit for what reason they are being written, and should be evaluated in this light. For real-world security, demonstrating attacks on fuzzy, ill-defined problems can be valuable when the potential harm is clear and immediate. For instance, it is valuable to show that language models can be manipulated to produce harmful content, even if we cannot precisely quantify "harmfulness". And when the objective is to advancing scientific understanding, we believe it is more productive to identify and focus on formal, well-defined sub-problems that can be rigorously studied, similar to how $\ell_p$-bounded perturbations provided a concrete framework for studying adversarial examples.

We acknowledge that even these well-defined sub-problems might still be challenging, just as achieving reliable $\ell_p$ robustness remains an open problem despite a decade of research. However, what we can definitely say is that if we cannot make progress on carefully scoped, formal problems, we have little hope of addressing the broader, fuzzier challenges of language model security. Moreover, working on well-defined problems enables rigorous scientific investigation: we can properly measure progress, compare different approaches, and build upon previous results. Attempting to solve the entire space of attacks without rigor is neither scientific nor likely to be productive.

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
