# OpenReview forum: "Position: Adversarial ML for LLMs Is Not Making Any Progress"
_NeurIPS.cc/2025/Position_Paper_Track — Submitted to NeurIPS 2025 Position Paper Track_

### Official Review · Reviewer_dNag · 2025-08-03

**Significance:** 4
**Presentation:** 4
**Rating:** 7
**Confidence:** 2

**Summary:**

The paper argues that, despite a decade of research into securing ML models, the shift to large language models has made adversarial problems (1) less clearly defined, (2) harder to solve, and (3) more difficult to evaluate—so much so that another decade of work risks yielding little meaningful advance. After reviewing how early work on bounded‐perturbation attacks and defenses struggled even on simple vision and classification tasks, the authors identify three core challenges in the LLM era: defining success without a single “task,” bounding an unbounded attack space, and ensuring reproducible, human‐centered evaluation of harm and utility. They illustrate these issues through six case studies—jailbreaks, un‐finetunable models, poisoning/backdoors, prompt injections, membership inference, and machine unlearning—showing that ad-hoc, human-driven attacks now often outperform automated methods.

**Strengths:**

1. To back up their position, the paper introduces a tabular framework (Table 1) that maps six representative research areas (e.g., jailbreaks, backdoors, membership inference) against the three challenge dimensions—defining, solving, evaluating—highlighting which sub‐challenges arise in each case. Section 3’s six detailed case studies then illustrate concrete examples (e.g., “LLM‐as‐judge” circularity in jailbreaks, unbounded threat models for prompt injections), grounding abstract claims in real‐world attack and defense scenarios.
2. A critical, reflective examination of the field’s ability to make measurable progress on adversarial robustness is both timely and important.
3. I particularly appreciate the alternative perspective discussion, which highlights how adversarial ML has evolved from an academic exercise—applying optimization and statistical theory to “toy” problems—into a top‐priority discipline in response to the large‐scale deployment of universal models in real‐world settings.

**Weaknesses:**

1. Overly provocative title, framing the topic as “not making any progress” risks dismissing valuable contributions in adversarial ML.
2. this paper doesn't present an real actionable roadmap on how we should progress to a research framework that closer to real-world challenges

**Questions:**

On a related note, does a "reverse scaling law" exist for adversarial robustness? I'm thinking of something analogous to μP for scaling laws, which would allow us to predict the adversarial behavior of large models by studying smaller ones.

**Alternative Position:**

Yes, and alternative positions are well-considered and addressed by the argument

**Author Identification:**

No.

**Context:**

4

**Discussion:**

4

**Ethics:**

["NO or VERY MINOR ethics concerns only"]

**Position:**

Yes, the paper argues for or against a position related to machine learning.

**Support:**

3

**Thoroughness:**

4

---

### Official Review · Reviewer_GRSi · 2025-08-12

**Significance:** 2
**Presentation:** 3
**Rating:** 6
**Confidence:** 4

**Summary:**

The paper argues that adversarial ML for LLMs is making little meaningful progress, as the shift from narrow, well-defined tasks to open-ended, general-purpose systems has made adversarial problems harder to define, solve, and evaluate.

**Strengths:**

1. The authors present a clear and assertive position that adversarial ML for LLMs is making limited progress, prompting the research community to critically examine its current direction and stimulating discussion.
2. The paper organizes difficulties into three coherent dimensions: defining, solving, and evaluating, each with concrete sub-challenges.
3. These six areas are central to current LLM safety debates and warrant focused attention.

**Weaknesses:**

1. Although it mentions alternative views, it does not substantively address cases where meaningful advances in adversarial ML for LLMs may already be occurring, which could make the argument appear one-sided.
2. Although the authors advocate focusing on well-defined sub-problems, they provide few concrete examples or detailed guidance on how to implement this recommendation.

**Questions:**

See weaknesses.

**Alternative Position:**

No

**Author Identification:**

No.

**Context:**

3

**Discussion:**

3

**Ethics:**

["NO or VERY MINOR ethics concerns only"]

**Position:**

Yes, the paper argues for or against a position related to machine learning.

**Support:**

3

**Thoroughness:**

5

---

### Official Review · Reviewer_y2jv · 2025-08-26

**Significance:** 4
**Presentation:** 4
**Rating:** 6
**Confidence:** 4

**Summary:**

This paper argues that adversarial machine learning research for large language models (LLMs) is facing fundamental challenges and slow progress. The authors highlight that defining, solving, and evaluating adversarial robustness for LLMs is much harder than for traditional models. Through several case studies, the paper shows that most current defenses are ad-hoc and poorly reproducible, and call for well-defined, scientifically rigorous problems to enable meaningful progress.

**Strengths:**

1. The paper is well-structured: it first points out the challenges in section 2, then uses relevant cases in section 3 to support the argument that adversarial ML needs well-defined problems. Finally, the authors offer their suggestions and return to their core argument.
2. This paper focuses on a fundamental problem in adversarial ML for LLMs.
3. The authors present a broad and well-chosen set of cases which precisely illustrate the impact and limitations caused by poorly defined research problems.

**Weaknesses:**

1. I agree with the authors that adversarial ML for LLMs is facing crucial problems with ill-defined challenges or objectives. However, the statement "is not Making Any Progress" is too extreme. Research is an exploratory process, and it requires a great deal of effort to reach a consensus on the definition of the problem. Challenges such as unbounded input or difficulty in evaluation are common and remain extremely challenging across research fields related to LLMs. This call itself is valuable, but previous works have made similar efforts.

2. In section 3, the authors point out many problems across different cases, but do not provide thoughtful solutions or insightful summaries. The suggestions in Section 4 are also rather high-level and fail to clearly demonstrate the authors' depth of thinking.

**Questions:**

It is suggested that similar problems occurring in different cases be described consistently, and that a summary of these common issues be provided at the beginning of section 3. For example, jailbreaks and prompt injections both face similar "unbounded adversary" problems, but they are described differently in the text.

**Alternative Position:**

Yes, and alternative positions are well-considered and addressed by the argument

**Author Identification:**

No.

**Context:**

3

**Discussion:**

3

**Ethics:**

["NO or VERY MINOR ethics concerns only"]

**Position:**

Yes, the paper argues for or against a position related to machine learning.

**Support:**

3

**Thoroughness:**

5

---

### Meta-Review · Area_Chair_UXXe · 2025-09-16

**Rating:** 7
**Confidence:** 4

**Strengths:**

Very timely paper. Strongly agree with the position.

**Weaknesses:**

As noted by reviewers, no deep insights are given about possible "Why"

**Questions:**

Adversarial Learning has been a failure but how come it is not a show stopper ?

**Thoroughness:**

3

---

### Decision · Program_Chairs · 2025-09-26

Reject